# Hygienic Characteristics and Detection of Antibiotic Resistance Genes in Crickets (*Acheta domesticus*) Breed for Flour Production

**Luca Grispoldi** [1,*] , **Musafiri Karama** [2] , **Saeed El-Ashram** [3,4] , **Cristina Maria Saraiva** [5,6] , **Juan García-Díez** [5] , **Athanasios Chalias** [1,7] , **Salvatore Barbera** [8] and **Beniamino T. Cenci-Goga** [1,2]

1   Medicina Veterinaria, Laboratorio di Ispezione degli Alimenti di Origine Animale, Università degli Studi di Perugia, 06126 Perugia, Italy; a.chalias@gmail.com (A.C.); beniamino.cencigoga@unipg.it (B.T.C.-G.)
2   Department of Paraclinical Sciences, Faculty of Veterinary Science, University of Pretoria, Onderstepoort 0110, South Africa; musafiri.karama@up.ac.za
3   School of Life Science and Engineering, Foshan University, Foshan 528231, China; saeed_elashram@yahoo.com
4   Faculty of Science, Kafr El-Shaikh University, Kafr El-Shaikh 33516, Egypt
5   Veterinary and Animal Research Centre (CECAV), University of Trás-os-Montes e Alto Douro, 5001-801 Vila Real, Portugal; crisarai@utad.pt (C.M.S.); juangarciadiez@utad.pt (J.G.-D.)
6   Department of Veterinary Sciences, School of Agrarian and Veterinary Sciences, University of Trás-os-Montes e Alto Douro, 5001-801 Vila Real, Portugal
7   European Food Safety Authority, EU-FORA Programme, 43126 Parma, Italy
8   Department of Agricultural, Forest and Food Sciences—AGRIFORFOOD, University of Turin, 10095 Grugliasco (TO), Italy; salvatore.barbera@unito.it
*   Correspondence: grisluca@outlook.it; Tel.: +39-075-585-7973

**Abstract:** During the last ten years, the worldwide interest in using insects as food and feed has surged. Edible insects fall within the category of novel foods, i.e., the category of food not consumed in significant amounts in the European Union before 15 May 1997 (the date of entry into force of Regulation (EC) No. 258/1997, later repealed by Regulation (EU) No. 2283/2015). One of the most promising insect species to be raised for food is the house cricket (*Acheta domesticus*). In this study, the rearing of a stock of house crickets was studied over a period of four months. The microbiological quality of the farm was studied using swabs on the surface of the rearing boxes to analyze the trend over time of different populations of microorganisms (total aerobic mesophilic microbiota, *Lactobacillus* spp., enterococci, *Staphylococcus* spp., *Enterobacteriaceae*, total coliforms, *Pseudomonas* spp. and molds). The presence of four antimicrobial resistance genes (*aph*, *blaZ*, *sul1*, and *tetM*) was investigated by polymerase chain reaction. A production scheme was also developed in order to obtain a cricket-based flour, which was analyzed for its microbiological and chemical-centesimal profile. The results obtained in this study demonstrate that the contamination increases with time and that a proper management of the farming system for insects is of the utmost importance, as it is for conventional farm animals such as ungulates, poultry, and rabbits. The old-fashioned adage "all full, all empty" for the farming system summarizes the need for proper cleaning and disinfection of the structures at the end of each production cycle.

**Keywords:** house cricket; entomophagy; novel foods; food safety; HACCP

## 1. Introduction

In the face of the continual evolution of society, food habits and the variety of tastes commonly available are frequently replaced by innovative products or products from foreign cultures on the consumer's table. Considering both the increase in population expected within 2050, reaching 9 billion people, and the parallel increase in the demand for proteins of animal origin, new protein sources have been suggested, including edible insects [1].

Edible insects fall within the category of novel foods, i.e., the category of food not consumed in significant amounts in the European Union before 15 May 1997 (the date of entry into force of Regulation (EC) No. 258/1997, later repealed by Regulation (EU) No. 2283/2015) [2]. A risk analysis is conducted for each novel food entering the European market and each new product or ingredient is subjected to the scientific opinion of EFSA (European Food Safety Authority) and requires formal authorization from the European Commission. These unconventional foods also have to guarantee they comply with the parameters of health and hygiene, safety, and quality, even in the absence of specific legislation.

The practice of consuming insects (or arthropods) is known as entomophagy, a term that first appeared in the English language in 1871. Although this is a little-known practice in western countries [3], it is familiar for over two billion people in the world. There is a list of more of 2000 edible species, which has been updated by Wageningen University and is continually on the increase [4]. Since 2012, the Food and Agriculture Organisation uses the term to focus attention on this topic, although the large variety of characteristics and species involved lead us to believe more specific terms will be coined in future. In some parts of the world, they are considered a natural source of sustenance and are naturally harvested in some places (e.g., Cambodia), whereas in others they constitute a delicacy (e.g., the *chapulines*—fried grasshoppers in Mexico) and some countries in Europe currently dispense them in the form of protein bars or in Michelin starred restaurants. The consumer is often unaware he is eating insects, as in the case of the colorant (E120) derived from *Dactylopius coccus* or cochineal.

So far, applications to EFSA for the following species have been submitted: *Hermetia illucens* (black soldier fly), *Alphitobius diaperinus* (lesser mealworm larvae), *Acheta domesticus* (house cricket), *Gryllodes sigillatus* (cricket), *Locusta migratoria* (locust), and *Tenebrio molitor* (yellow mealworm). Recently, *Tenebrio molitor* larvae have been considered by EFSA as fit for human consumption [5]. EFSA has also produced a scientific opinion on the risk profile related to the production and consumption of insects as food and feed [6]. The opinion, in the form of a risk profile, presents potential biological and chemical hazards as well as allergenicity and environmental hazards associated with farmed insects used as food and feed taking into account the entire chain, from farming to the final product.

From a nutritional point of view, insects represent an interesting source of nutrients such as vitamins, mineral salts, and proteins in particular [7,8]. Insects are also considered to have excellent food conversion rates (how much feed must be provided for each kg of food produced), higher relative growth, and lower greenhouse gas emissions when compared with pigs and cattle [9,10].

As a result, interest in the possibility of breeding insects for food has increased in recent years. The purpose of this study was to study a prototype of a cricket (*Acheta domesticus*) farm for food purposes, with particular attention to the microbiological characteristics of the rearing environment, the presence of genes encoding for antibiotic resistance and the characteristics of the final product (cricket flour). Considerations have been made regarding animal welfare, in particular, to avoid overpopulation and consequent cannibalism phenomena and to avoid suffering at the time of slaughter.

## 2. Materials and Methods

### 2.1. Crickets Rearing

The rearing of a stock of house crickets (*Acheta domesticus*) was studied over a period of four months. The rearing temperature was kept constant at around 26 °C in a conditioned chamber. Temperature and humidity during the entire period were recorded using a data-logger. The crickets (500 adults per box) were reared in plastic containers of 70 cm × 40 cm × 40 cm equipped with a special lid to prevent the escape of the insects. Considering a crawl space of 2800 cm$^2$ in each box, we had 1 cricket per 5.6 cm$^2$. This density of population was chosen to avoid overpopulation and consequent cannibalism phenomena that greatly increase when crowding exceeds 1 cricket per 2.5 cm$^2$ [11]. Peat

was used both as litter and for the ovipositional trays. The diet consisted of bran, vegetables, and fresh fruit (mainly apple). Water was provided via soaked sponges changed every other day. Eggs were collected once a week and the ovipositional trays (with 3–4 cm of peat) were kept at a temperature of 30 °C, and the humidity was adjusted between 70–75% twice a day using a water vaporizer. Eggs hatched in two weeks. Nymphs were reared apart for about two weeks (the amount of time needed to complete the first three molts and to reach a length of 1.1–1.3 cm). Then, they were moved to the boxes with the other subjects. The number of molts in the course of development varied from 6 to 12 (about 3-day intervals between each other).

## 2.2. Microbiological Analysis

Once a week, sterile swabs were used to collect samples from a total of nine rearing boxes on a surface of 100 cm$^2$ each. Samples were transported to the laboratory in a refrigerated container. Tenfold dilutions were prepared in sterile tubes with 9 mL of Maximum Recovery Diluent (MRD, Oxoid, Basingstoke, Hampshire, UK). Dilutions were inoculated in triplicate on different culture media. The total aerobic and mesophilic microbiota was determined on Plate Count Agar (PCA; Oxoid) at 30 °C for 72 h; *Lactobacillus* spp. on Man, Rogosa and Sharpe Agar (MRS; Oxoid) pH 5.5, at 30 °C for 72 h under anaerobic conditions (Gas generating kit, Oxoid); enterococci on enterococcus agar (ENT; Oxoid), at 37 °C for 48 h; *Staphylococcus* spp. on Baird Parker agar (BP; Oxoid) containing Egg Yolk Tellurite (Oxoid) at 37 °C for 48 h; *Enterobacteriaceae* on violet red bile glucose agar (VRBG; Oxoid) at 37 °C for 24 h; total coliforms on violet red bile lactose agar (VRBL; Oxoid) at 37 °C for 24 h; *Pseudomonas* spp. on pseudomonas agar base (PS103; Oxoid) at 37 °C for 24 h; molds on Chloramphenicol Yeast Glucose Agar (CYG; HiMedia) at 25 °C for 72 h. The colonies were then counted on all the plates, using a colony count viewer (Petri light, PBI, Milan) and colony counter pen (Colony Count, PBI, Milan). All values were converted into logs and the arithmetic mean was calculated for each sampling. Samples were then divided into two groups: samples from day 0 to day 28 were grouped in the "first cycle" group, while samples from day 42 to day 63 were grouped in the "second cycle" group. This division was chosen because the period between day 28 and day 42 corresponds to the addition of the young crickets to the rearing boxes. Statistical analyses were performed with StatView 5.0.1 for Mac OS (SAS Institute, Inc., Cary, NC, USA). Unpaired comparison by unpaired t-test was performed to determine if the likelihood of observed differences between the two groups (bacterial counts for the first cycle, and bacterial counts for the second cycle) occurred by chance. The chances are reported as *p* values which are given in the box plots for each microbial group.

## 2.3. Antimicrobial Resistance Genes Research by PCR

Naturally dead crickets were taken from the nine different rearing boxes to check for the presence of antimicrobial resistance genes. Samples were frozen in liquid nitrogen and ground in order to obtain a homogeneous pulverized pool. The HipurA™ Insect DNA Purification Kit from the HiMedia company (Mumbai, India) was used to extract the DNA. The quantification of the extracted genetic material was performed using the NanoDrop™ Lite spectrophotometer (Thermo Fisher Scientific, Waltham, Massachusetts, USA) with 1 μL of sample. The DNA amplification was conducted on a volume of 25 μL using 12.5 μL of RED Taq (10 mM Tris HCl, pH 8.3, 50 mM KCl, 1.5 mMMg Cl2, 0.001% of gelatin, 0.2 m Meach of deoxyribonucleoside triphosphate), 0.5 μL (1 μM) of each primer, 5 μL of extracted DNA and 6.5 μL of H2O. The PCR reaction was carried out in a thermocycler Gene Amp, PCR System, 9700 Gold (Applied Biosystems, Foster City, CA, USA). The primers and the amplification conditions used are listed in Table 1. The amplifications were analyzed by an electrophoretic run on 1.5% agarose gel containing ethidium bromide (0.5 μg/mL); 10 μL of each PCR sample was loaded with 2 μL of 6× loading buffer (Fermentas-VWR-Italy) and 5 μL of marker PCR as reference DNA (Fermentas-VWR-Italy); the run was carried out at a voltage of 100 V for about 1 h in TBE 10× (Trizma base, boric acid, EDTA 0.5 MpH

8). At the end of the run, the bands were viewed with the UV transilluminator (Fotodine 3–3102 Celbio, Milan, Italy).

**Table 1.** Primers and the amplification conditions used.

| Target Gene | Description | Nucleotide Sequence (5'-3') | Ampl (bp) | Amplification |
|---|---|---|---|---|
| *aph* | *aac(6′)-aph(2″)* gene, which encodes for a bi-functional, aminoglycoside modifying enzyme [12] | GAGCAATAAGGGCATACCAAAAATC CCGTGCATTTGTCTTAAAAAACTGG | 505 bp | 94 °C × 5′; (94 °C × 30″, 55 °C × 30″, 72 × 30″) × 35 cycles, 72 °C × 7′ |
| *blaZ* | *blaZ* gene, which encodes for the predominantly β-lactamase in S. aureus [13] | ACTTCAACACCTGCTGCTTTC TGACCACTTTTATCAGCAACC | 173 bp | 94 °C × 4′; (94 °C × 30″, 58 °C × 30″, 72 × 30″) × 30 cycles, 72 °C × 7′ |
| *tetM* | *tetM* gene, which encodes for a tetracycline resistance protein [14] | ACCCGTATACTATTTCATGCACT CCTTCCATAACCGCATTTTG | 1115 bp | 95 °C × 3′; (95 °C × 1′, 48 °C × 1′, 72 × 1′) × 35 cycles, 72 °C × 10′ |
| *sul1* | *sul1* gene normally found in class 1 integrons, which encodes for a form of dihydropteroate synthase responsible for sulphonamide resistance in gram-negative bacilli [15] | CGGCGTGGGCTACCTGAACG GCCGATCGCGTGAAGTTCCG | 433 bp | 94 °C × 3′; (94 °C × 15″, 69 °C × 30″, 72 × 1′) × 30 cycles, 72 °C × 7′ |

*2.4. Flour Production*

The following production scheme has been developed in order to obtain a flour-based on crickets: 24 h before the start of the production process, the food is removed from the rearing boxes in such a way as to facilitate insect purging. After this period, crickets were collected from the various boxes and transported to the laboratory. They were put into a bag that was seal closed and placed in the freezer at −20 °C for 24 h. Insects are ectothermic, which means that in cold temperatures, their metabolism slows down until death. The insects go into a cold-induced coma from which they do not recover, so there is no violent death or change in state [16]. The first step to be carried out in the laboratory is washing. The crickets undergo three washes in running water. Then, they are weighed and pasteurized in boiling water for about 5 min. This step is essential to decrease the microbial load present on the surface of the insects. Once the pasteurization time has elapsed, they are placed in a dryer (for the tests carried out, a dryer that worked at a temperature of 55 °C was used), distributed evenly over the entire surface of the plate, overnight. After the time necessary to obtain an adequate weight loss, all parts of the cricket were ground into flour using a mortar and pestle. The flour obtained was dark green/straw yellow in color, with a sweet smell, similar to hazelnut but slightly acrid, was packaged in conditions of absolute sterility with the aim of subsequently subjecting it to microbiological and chemical-centesimal analyses.

**3. Results**

The temperature and relative humidity of the rearing room during the entire period were recorded by the chamber data-logger and are reported in Figure 1. The mean temperature was 27.7 ± 2.4 °C and the mean relative humidity was 45 ± 8.5%.

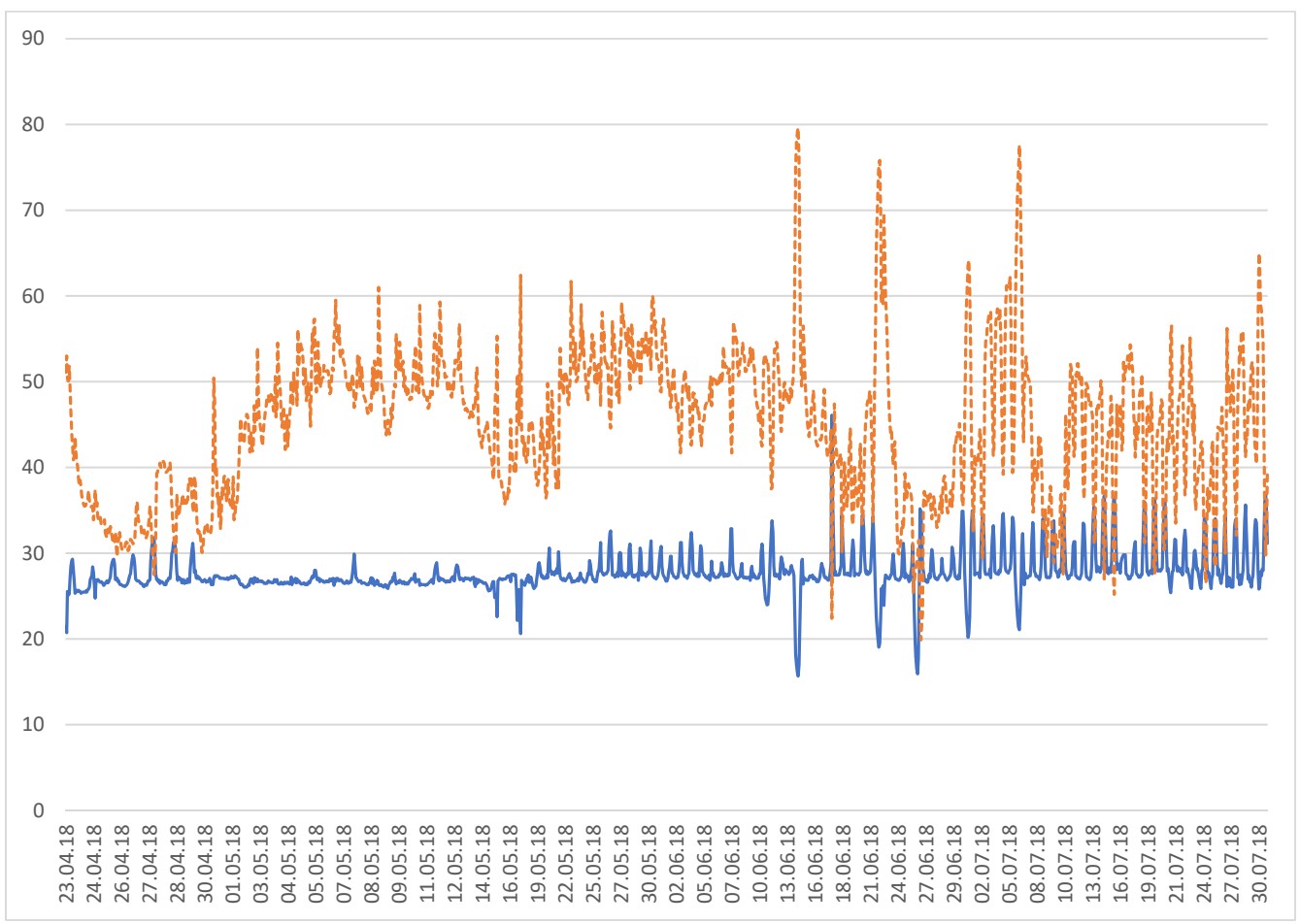

**Figure 1.** Temperature (°C, continuous line) and relative humidity (%, dashed line) of the rearing room.

The trend over time of the means calculated for the different bacterial populations (expressed as log cfu/cm$^2$) in the rearing boxes is shown in Table 2. The total aerobic mesophilic count was $4.44 \pm 1.14$ cfu/cm$^2$ at t0 and reached $5.38 \pm 0.28$ cfu/cm$^2$ at t63, with its maximum at t49 with a concentration of $5.61 \pm 1.06$ cfu/cm$^2$. *Pseudomonas* spp. count was $3.63 \pm 1.44$ cfu/cm$^2$ at t0 and reached $3.71 \pm 0.89$ cfu/cm$^2$ at t63, with its maximum at t49 with a concentration of $4.92 \pm 0.45$ cfu/cm$^2$. The *Enterobacteriaceae* count was $3.49 \pm 0.64$ cfu/cm$^2$ at t0 and reached $4.68 \pm 0.33$ cfu/cm$^2$ at t63, with its maximum at t49 with a concentration of $5.32 \pm 0.55$ cfu/cm$^2$. The total coliforms count was $3.56 \pm 0.66$ cfu/cm$^2$ at t0 and reached $4.38 \pm 0.81$ cfu/cm$^2$ at t63, with its maximum at t49 with a concentration of $4.66 \pm 0.51$ cfu/cm$^2$. The enterococci count was $4.28 \pm 0.82$ cfu/cm$^2$ at t0 and reached $4.60 \pm 0.68$ cfu/cm$^2$ at t56, with its maximum at t28 with a concentration of $5.49 \pm 0.34$ cfu/cm$^2$. *Lactobacillus* spp. count was $4.70 \pm 0.97$ cfu/cm$^2$ at t0 and reached $5.08 \pm 0.53$ cfu/cm$^2$ at t63, with its maximum at t49 with a concentration of $5.86 \pm 1.12$ cfu/cm$^2$. *Staphylococcus* spp. count was $3.74 \pm 0.99$ cfu/cm$^2$ at t0 and reached $4.77 \pm 0.46$ cfu/cm$^2$ at t63, which was also its maximum concentration. The mold count was $2.06 \pm 1.24$ cfu/cm$^2$ at t0 and reached $3.14 \pm 0.32$ cfu/cm$^2$ at t63, which was also its maximum concentration. *p*-values for total aerobic mesophilic microbiota, lactobacilli, *Enterobacteriaceae*, total coliforms, staphylococci, and enterococci showed that counts were significantly higher in the second cycle (Figures 2–4).

**Table 2.** The trend over time of the means calculated for the different bacterial populations (expressed as log cfu/cm$^2$) on the surface of the rearing boxes.

| | PCA | | PS 103 | | VRBL | | VRBG | | ENT | | MRS | | BP | | CYG | |
|---|---|---|---|---|---|---|---|---|---|---|---|---|---|---|---|---|
| | Mean | s.d. | Mean | s.d. | Mean | s.d. | Mean | s.d. | Mean | s.d. | Mean | s.d. | Mean | s.d. | Mean | s.d. |
| T0 | 4.44 | 1.14 | 3.63 | 1.44 | 3.56 | 0.66 | 3.49 | 0.64 | 4.28 | 0.82 | 4.70 | 0.97 | 3.74 | 0.99 | 2.06 | 1.24 |
| T7 | 3.71 | 0.78 | 2.10 | 0.91 | 2.38 | 1.21 | 2.48 | 1.16 | 3.74 | 1.18 | 4.14 | 0.98 | 2.14 | 0.59 | 1.50 | 0.71 |
| T14 | 3.90 | 1.58 | 3.30 | 0.71 | 3.13 | 0.83 | 3.29 | 0.92 | 3.56 | 1.06 | 4.39 | 0.89 | 3.33 | 0.98 | 2.00 | |
| T21 | 3.71 | 1.03 | 4.48 | | 3.10 | 1.56 | 2.80 | 1.30 | 2.83 | 0.34 | 3.60 | 1.34 | 2.50 | 0.58 | 2.00 | |
| T28 | 4.59 | 1.50 | 2.00 | | 2.34 | 0.31 | 2.41 | 0.35 | 5.49 | 0.34 | 5.21 | 0.53 | 1.23 | 0.40 | 1.89 | 0.16 |
| T42 | 4.90 | 0.62 | 3.73 | 0.53 | 3.77 | 0.91 | 4.38 | 0.59 | | | 5.05 | 0.47 | 3.97 | 0.31 | | |
| T49 | 5.61 | 1.06 | 4.92 | 0.45 | 4.66 | 0.51 | 5.32 | 0.55 | 5.40 | 0.93 | 5.86 | 1.12 | 3.91 | 0.72 | | |
| T56 | 4.38 | 1.46 | 2.68 | 0.88 | 2.35 | 0.78 | 2.64 | 0.83 | 4.60 | 0.68 | 4.38 | 0.96 | 4.25 | 1.43 | | |
| T63 | 5.38 | 0.28 | 3.71 | 0.89 | 4.38 | 0.81 | 4.68 | 0.33 | | | 5.08 | 0.53 | 4.77 | 0.46 | 3.14 | 0.32 |

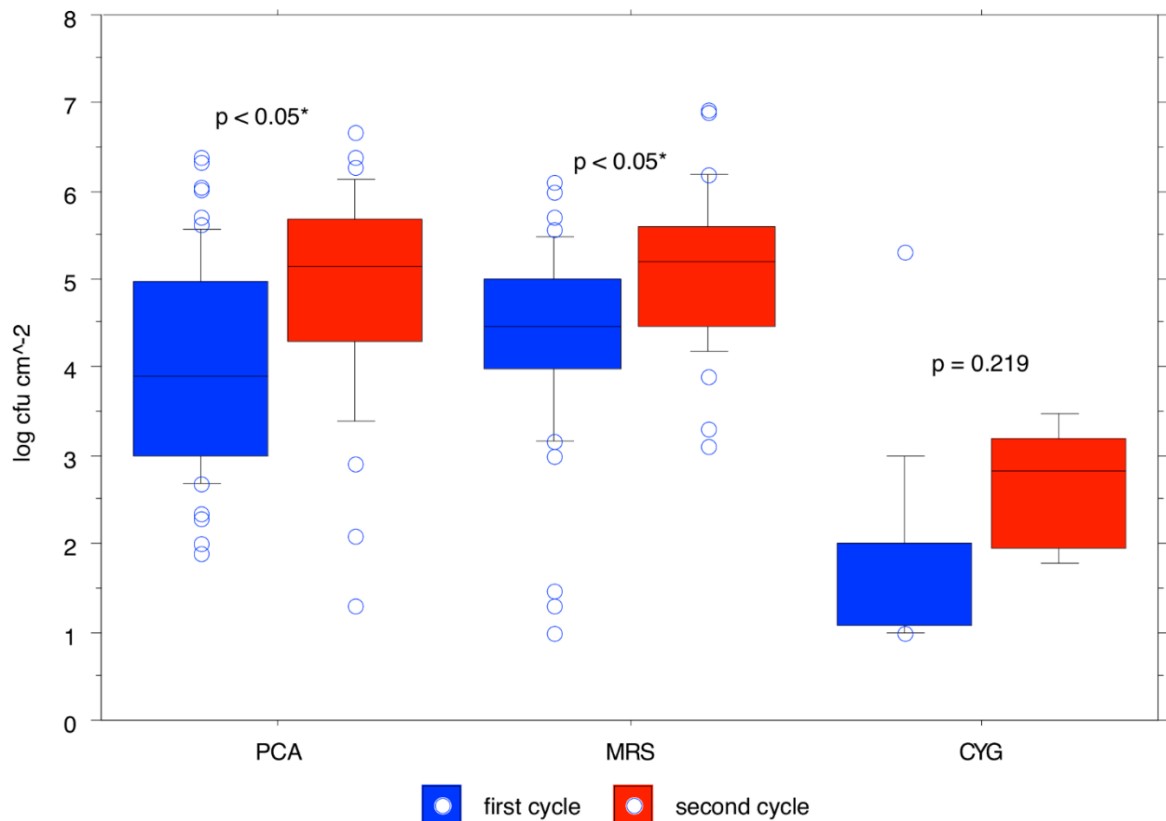

**Figure 2.** The likelihood of observed differences between bacterial counts for the first cycle and bacterial counts for the second cycle for total aerobic mesophilic microbiota, *Lactobacillus* spp. and molds.

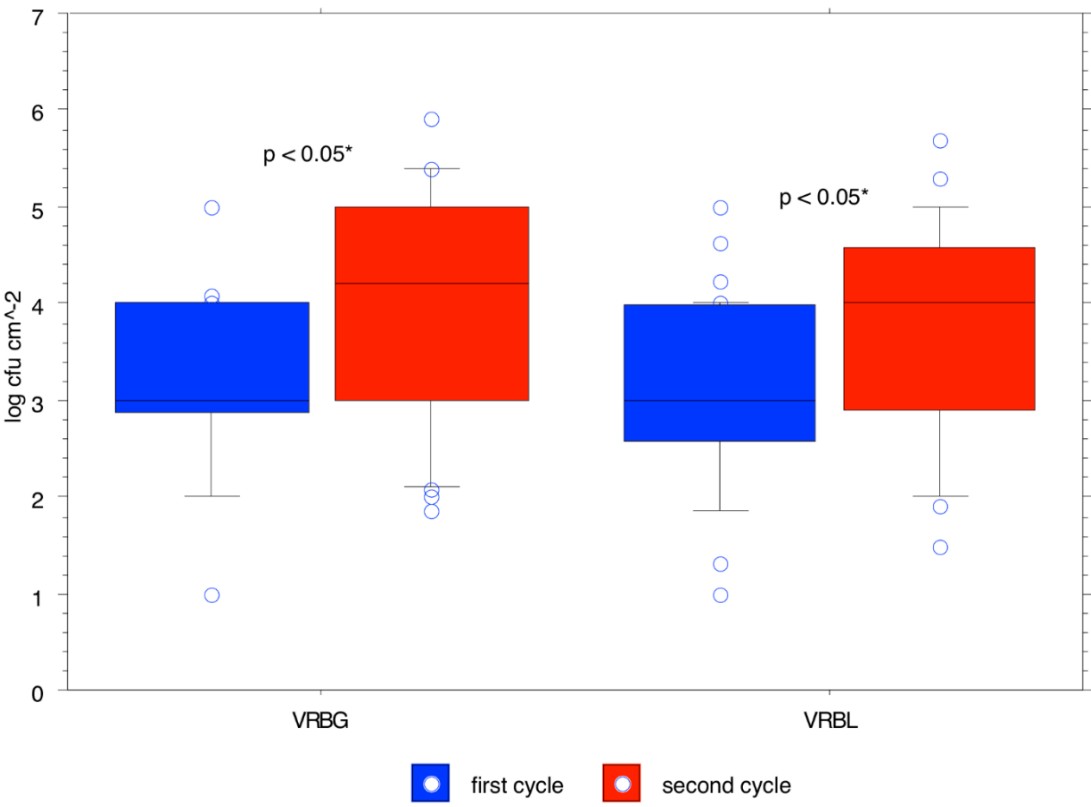

**Figure 3.** Likelihood of observed differences between bacterial counts for the first cycle and bacterial counts for the second cycle for *Enterobacteriaceae* and total coliforms.

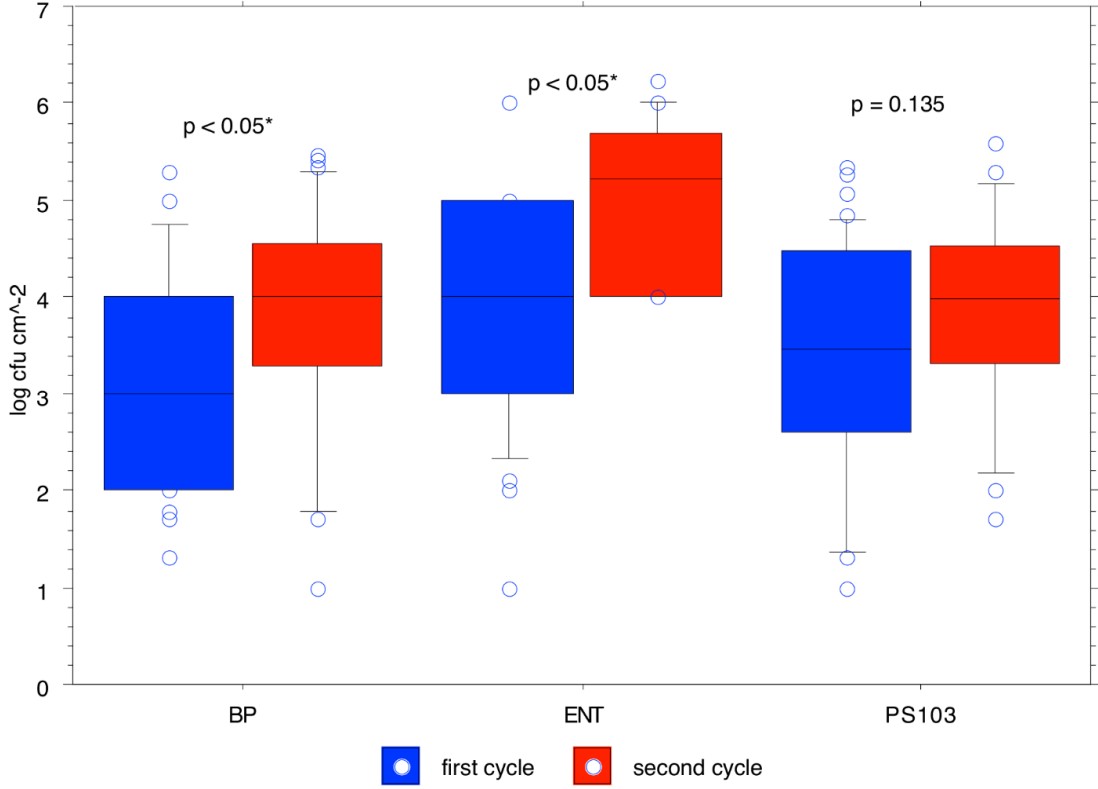

**Figure 4.** Likelihood of observed differences between bacterial counts for the first cycle and bacterial counts for the second cycle for *Staphylococcus* spp., enterococci, and *Pseudomonas* spp.

The research by PCR of the genes that encode for antimicrobial resistance showed no positives for *aph*, *blaZ*, and *sul1*. Instead, four out of nine samples were positive for the presence of the *tetM* gene.

The chemical-centesimal analyses of the flour showed the following results: carbohydrates 9.5 g/100 g, ashes 3.1 g/100 g, dietary fiber <0.5 g/100 g, total fat 15 g/100 g, saturated fats 4.4 g/100 g, proteins 58.9 g/100 g, NaCl 0.61 g/100 g, humidity 13.5%, and energy value 408.6 kcal/100 g. The microbiological analysis of the flour did not highlight the presence of bacterial populations of interest within the detection limits of the methodology used.

## 4. Discussion

Currently, European legislation does not provide specific microbiological criteria for whole insects or insect products for human consumption. We decided to use the bacterial load of the surface as a process hygiene criterion and the bacterial load of the cricket flour as a food safety criterion. Some authors suggested using the total aerobic bacterial count provided by the European Commission Regulation (EC) No. 2073/2005 [17] for ground beef as a guideline for food safety and final product hygiene values [18]. The total aerobic bacterial counts for crickets reported in the literature vary in a range from $10^4$ cfu/g to $10^8$ cfu/g. This high variability reflects the differences in the protocol used for rearing and processing the insects before the transformation in the final product [19]. Considering that the whole animals are used to produce food, including their gut, a common practice also applied by the authors in the present work was to decrease the microbial loads by applying fasting 24–48 h before the kill step. If not applied, the microbial load for whole crickets reported in the literature is much higher (up to $10^{12}$ cfu/g) [20] if confronted with the limits provided by the law. In this study, we decided to focus on the microbial load of the surfaces of the rearing boxes. If compared with the limits provided by the EC Regulation 2073/2005 for the total aerobic count and *Enterobacteriaceae* count on the surface of the carcasses of different species, the results obtained in this study are interesting. The total aerobic count was lower than the upper limit provided (5.0 log cfu/cm$^2$) in seven out of nine of the time analyzed. The level of contamination was higher for *Enterobacteriaceae*, which were compliant only in four out of nine cases. Results obtained during this study confirm the high microbial diversity in crickets, as reported by other authors [21,22]. The presence of fungal species (molds in particular) has been reported both by breeders in insect-farming facilities and in rearing experiments at the Swedish University of Agricultural Sciences (SLU) without involving any major mortality [23]. Other authors reported that yeast and mold counts for crickets were above the Good Manufacturing Practice (GMP) limits for raw meat [18,19]. The statistical analysis demonstrated that counts were significantly higher in the second cycle for total aerobic mesophilic microbiota, lactobacilli, *Enterobacteriaceae*, total coliforms, staphylococci, and enterococci if compared to the first cycle. This is due to the continued presence of the animals (of all ages and growth stages) in the boxes themselves. This suggests that an "all full, all empty" approach with disinfection of the rearing environment is advisable to avoid excessive and unwanted increases in the bacterial load.

Many insect species are known as vectors for bacteria that bring genes encoding for antimicrobial resistance [24,25]. Our results are consistent with a study by Milanovic et al. [26], which investigated the presence of antimicrobial resistance genes in edible insects by using both classic- and nested-polymerase chain reaction and reported the presence of tetracycline resistance genes in cricket samples.

Regarding the hygienic characteristics of the cricket flour produced as described above, at a legislative level, there are no specific items that provide precise parameters. If we compare the results obtained in this study with the parameters provided by the regulations for cereal and mixed flours, we can note that the values of *Enterobacteriaceae* (which must be less than 1000 cfu/g) and *Salmonella* (absent in 25 g) are widely respected. Likewise, cricket flour met the parameters set for powdered milk and powdered whey. The fact that the

microbiological analyses carried out on cricket flour showed that the bacterial populations was reduced by pasteurization below the detection limit of the method used suggests that this product does not present any harm for food safety if correctly handled and stored.

## 5. Conclusions

During the last ten years, the worldwide interest in using insects as food and feed surged. One of the most promising insect species to be raised for food is the house cricket (*Acheta domesticus*). Our study is, to the best of our knowledge, the first organic approach for the definition of (i) environmental contamination level for farmed insects, (ii) insect load for animal welfare and prevention of cannibalism and contamination, and (iii) animal stunning for animal welfare consideration.

Our study, therefore, proved that the contamination increases with time and that proper management of the farming system for insects is of the utmost importance, as it is for conventional farm animals such as ungulates, poultry, and rabbits. The old-fashioned adage "all full, all empty" for the farming system summarizes the need for proper cleaning and disinfection of the structures at the end of each production cycle. Moreover, now that insects have been authorized for sale in many countries, it is mandatory to have available data and know-how for food safety of insects. In Europe, *Tenebrio molitor* larvae have been considered by EFSA as fit for human consumption [5].

In recent years there has been a plethora of papers describing microbiota, and microbial diversity of edible insects by metagenomic sequencing [27–29]. These works, while interesting from a zoological and entomologic point of view, are deceiving when used as tools for food safety. Insects can carry over the environmental contamination and act either as a reservoir or, very probably, as vectors for human pathogens. We do strongly believe that the correct approach for the definition of food safety standard of insects for human consumption is similar to any HACCP (Hazard Analysis and Critical Control Point), GMP (Good Manufacturing), and the more recent HARPC (Hazard Analysis and Risk-based Preventive Controls) introduced by the US FDA (Food and Drug Administration) with the FSMA (Food Safety Modernization Act).

**Author Contributions:** Conceptualization, B.T.C.-G. and L.G.; methodology, B.T.C.-G. and L.G.; formal analysis, B.T.C.-G., L.G., and S.B.; writing—original draft preparation, L.G., B.T.C.-G., and A.C.; writing—review and editing, L.G., B.T.C.-G., M.K., and S.E.-A.; supervision, C.M.S. and J.G.-D.; project administration, B.T.C.-G.; funding acquisition, B.T.C.-G. All authors have read and agreed to the published version of the manuscript.

**Funding:** This research was funded by EFSA (European Food Safety Authority), grant number GP/EFSA/ENCO/2018/05_GA6.

**Conflicts of Interest:** The authors declare no conflict of interest.

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
