# Peer review of "Hygienic Characteristics and Detection of Antibiotic Resistance Genes in Crickets (Acheta domesticus) Breed for Flour Production"

_2036-7481, doi:10.3390/microbiolres12020034_

Round 1

Reviewer 1 Report

In the present study, the bacteria counts were significant higher in the second cycle, so the authors suggested that it was caused by adding new young crickets. There should be a control group. What if they kept the crickets without adding new young crickets?

Even the bacteria load on rearing boxes was high, there was no bacteria detected after the pasteurization. In this situation, is there any harm for food safety? Can author discuss about it?

In the present study, bacteria load of rearing boxes was detected. But the limits provided by EC Regulation 2073/2005 was about bacteria load on the surface of the carcasses. What’s the different meaning between this two? Why the authors only detected the bacteria load of rearing boxes?

Author Response

Response to Reviewer 1 Comments

Thank you very much for your comments and suggestion. You can find the changes in the manuscript highlighted in yellow.

Point 1: In the present study, the bacteria counts were significant higher in the second cycle, so the authors suggested that it was caused by adding new young crickets. There should be a control group. What if they kept the crickets without adding new young crickets?

Response 1: Considering the rapid growth rate of crickets under optimal temperature and feeding conditions (such as those used in this study) it is necessary to place the newborns in the boxes to maintain a potential continuous production of adults to be transformed into flour. In our study we do not suggest that the increase in bacterial populations in the second cycle is due to the placing of young specimens in the boxes, but that it is due to the continued presence of the animals (of all ages and growth stages) in the boxes themselves. For this reason, as described in lines 229-230 and 255-257, we suggest that a “all full, all empty” approach with disinfection of the rearing environment is advisable to avoid excessive and unwanted increases in the bacterial load.

Point 2: Even the bacteria load on rearing boxes was high, there was no bacteria detected after the pasteurization. In this situation, is there any harm for food safety? Can author discuss about it?

Response 2: The fact that the microbiological analyzes carried out on cricket flour showed that the bacterial populations have been reduced by pasteurization below the detection limit of the method used suggests that this product does not present any harm for food safety if correctly handled and stored. I have added this statement in the discussion section, at lines 241-245.

Point 3: In the present study, bacteria load of rearing boxes was detected. But the limits provided by EC Regulation 2073/2005 was about bacteria load on the surface of the carcasses. What’s the different meaning between this two? Why the authors only detected the bacteria load of rearing boxes?

Response 3: Considering that there are no specific items that provide precise parameters at a legislative level for edible insects and insects-based products, we decided to use the bacterial load of the surface as a process hygiene criteria and the bacterial load of the cricket flour as food safety criteria.

Reviewer 2 Report

I consider the work to be clear, concise, well scientifically processed and with a good methodology.
This aspect of edible insect breeding with a focus for practice has not been studied much so far, so it is great the authors chose this topic.

Author Response

Response to Reviewer 2 Comments

Point 1: I consider the work to be clear, concise, well scientifically processed and with a good methodology.
This aspect of edible insect breeding with a focus for practice has not been studied much so far, so it is great the authors chose this topic.

Response 1: Thank you very much for your comments.

Round 2

Reviewer 1 Report

I consider the author's reply can be accepted.